# Experimental Analysis of Steel Circular Hollow Section under Bending Loads: Comprehensive Study of Mechanical Performance

**DOI:** 10.3390/ma15124350

**Published:** 2022-06-20

**Authors:** Manahel Shahath Khalaf, Amer M. Ibrahim, Hadee Mohammed Najm, Mohanad Muayad Sabri Sabri, Samadhan Morkhade, Ashish Agarwal, Mohammed A. Alamir, Ibrahim M. Alarifi

**Affiliations:** 1Department of Civil Engineering, College of Engineering, University of Diyala, Baqubah 32001, Iraq; 2Department of Civil Engineering, Zakir Husain Engineering College, Aligarh Muslim University, Aligarh 202002, India; gk4071@myamu.ac.in; 3Peter the Great St. Petersburg Polytechnic University, 195251 St. Petersburg, Russia; mohanad.m.sabri@gmail.com; 4Department of Civil Engineering, Vidya Pratishthan’s Kamalnayan Bajaj Institute of Engineering & Technology (VPKBIET), Baramati, Pune 413133, India; samadhanmorkhade@gmail.com; 5Department of Civil Engineering, J.C. Bose University of Science and Technology, YMCA, Faridabad 121002, India; ashishagarwal.ce@gmail.com; 6Department of Mechanical Engineering, College of Engineering, Jazan University, Jazan 45142, Saudi Arabia; malamir@jazanu.edu.sa; 7Department of Mechanical and Industrial Engineering, College of Engineering, Majmaah University, Al-Majmaah, Riyadh 11952, Saudi Arabia; i.alarifi@mu.edu.sa

**Keywords:** bending, local buckling, steel tubes, circular hollow section, stiffness, bearing capacity, strength

## Abstract

The present study aimed at evaluating the mechanical performance under bending loads of circular hollow sections of steel. Different bending tests have been carried out by applying two-point loads, to determine and examine the effects of the diameter, the thickness of the section, and the span of the beam on the performance of the steel tube. The effects of square opening and variation in the number of openings on the performance of these sections have also been examined. Ten samples of hollow circular beams of varying thickness (2 mm, 3 mm, and 6 mm), varying diameter (76.2 mm, 101.6 mm, and 219 mm), and varying span (1000 mm, 1500 mm, and 2000 mm) were fabricated and tested for pre-failure and post-failure stages. The dimensions of the reference specimen considered were 3 mm in thickness, 101.6 mm in diameter, and 1500 mm in span. The results have shown that on increasing the section thickness by 200%, ductility and bearing strength were enhanced by 58.04% and 81.75%, respectively. Meanwhile, decreasing the section thickness by 67%, ductility and bearing strength were reduced by 64.86% and 38.87%, respectively. Moreover, on increasing the specimen diameter and on decreasing span, a significant increase in bearing strength and stiffness was observed; however, ductility was reduced. Meanwhile, on increasing the span of the specimen, all the parameters observed, i.e., bearing strength, stiffness, and ductility, decreased. On observing the ultimate strength of each specimen with square opening, the ultimate strength was reduced by 17.88%, 19.71%, and 14.23% for one, two-, and three-square openings, respectively. Moreover, the ductility was significantly reduced by 72.40%, 67.71%, and 60.88% for one, two-, and three-square openings/apertures, respectively, and led to the sudden failure of these specimens. The local buckling failure dominated for specimens having a D/t ratio more than 50 and showed very negligible levels of ovalization of the cross-section. Local buckling failure was observed to be prevented after providing the circular rings in the specimen, since bearing strength increased compared with the specimen without rings.

## 1. Introduction

Over the past few decades, steel has become one of the most significant and widespread structural materials, thus attracting many research efforts to investigate its sectional strength and structural behavior. In many cases, steel constructions are prepared as alternatives to reinforced concrete ones due to the following advantages [1]:a.It has high tensile strength compared to concrete;b.Structural steel has high strength, toughness, stiffness, and tensile properties;c.Steel can be developed into any shape, either welded or bolted together in construction;d.Steel construction is fast, which reduces the time required to construct the project, and it can be disassembled easily without losing the integrity of the structure.

Structural steel development has explored various avenues in terms of the diversity of steel sections to be employed as construction materials. Out of these, the hollow structural section is considered to be one of the most reliable sections due to its excellent properties.

In mechanical and construction applications, circular hollow sections (CHSs) are perhaps the most efficient and flexible structure. Therefore, they are used in various well-designed and strong steel structures built worldwide. Steel circular hollow sections (CHSs) increase buildings’ strength to weight ratio during serviceability; in turn, it reduces the use of materials and allows for a greater building span, enhancing the structural efficiency and reducing the cost [1].

High-strength welded steel tubes, utilized as structural components in buildings and other structures and a range of manufactured items, are referred to as hollow structural sections [2,3,4,5]. They are manufactured to ASTM A500, A1085, and A1065 requirements in round, square, and rectangular forms and a wide range of sizes [6]. A circular hollow steel tube has been shown to be a strong structural element for constructions, jibs, skyscrapers, cranes, barriers, and fluid mechanics equipment [7,8]. The above structural system has inherent efficiency because of its ideal compression, tension, torsion, and bi-axial bending properties [9]. However, the widespread usage of steel tubes in earthquake applications for the construction industry is restricted because of the incomplete understanding of their behavior after being subjected to bending loads and the probable absence of flexibility and steady behavior over several cycles of loads [9,10,11].

Construction applications of steel tubes have been confined to static loads on column members, truss components, bracing, and wall elements [12]. The advanced seismic frame uses, such as column and beam structural elements, can significantly benefit lower seismic weight, reduced lateral bracing, modular construction applications, and innovative rehabilitation procedures [9,13]. Due to the large percentage of inelastic damage occurring in the beam elements, existing seismic standard codes require a strong-column weak-beam mechanism during the design [14,15,16]. To achieve this, a better understanding of the bending behavior of these elements and an appropriate method of modeling this behavior are required before steel tubes can be used for seismic design applications [17,18,19].

Until the present time, most studies on steel tube flexural behavior have been limited to beam–column structural elements. Dywer and Galambos [20] studied various beam-column specimens under collapse, emphasizing the relevance of the ratio of axial load (P/Py) and the ratio of slenderness (L/r). The influence of the slenderness ratio and axial load ratio on the bending behavior was investigated by Nakashima and Liu [10] in order to obtain better knowledge of the plastic behavior of steel tube columns in seismic constructions. To explain the plastic behavior of a steel tube column under varied axial load conditions, Wang et al. [13] conducted hybrid experimental works. For both concrete-filled tubes (CFTs) and hollow sections, further investigations looked at the behavior connections between broad flange beams and steel tube columns [21,22]. Several experimental studies focused on the bending behavior of steel tube beam elements under a range of monotonic loading regimes. The aspect ratio (b/h), depth-to-thickness ratio (h/t), and width-to-thickness ratio (b/t) all had a role in these experiments [23,24]. The flexural behavior of steel tube beam members under bending stresses was studied in more recent experimental works [14,15]. The findings of these experimental works emphasize the relevance of the h/t and b/t found throughout experimental testing and give insights into the predicted local buckling behavior. Lateral constraint, bending moment variation, and overall member slenderness have influenced steel tube performance under repeated bending loads [23,25].

Reasons for using circular hollow sections are as follows. (a) Because of the homogeneous distribution of cross-section materials around the polar axis, CHSs have several structural properties under various loading circumstances, such as bending, axial, and torsion, among others. This distinguishes them in terms of performance. [26]. (b) CHSs are frequently employed to construct tall bridge structures since these assists in lowering their density and hence their self-weight. It also aims to reduce the size of foundations, saving money in construction [27]. (c) The CHS section possesses rounded edges and a minimal surface area as compared to other open sections, as shown in Figure 1, which reduce the likelihood of corrosion. Hence, because of the smooth change at the joints and closed section, this results in minimal corrosion protection costs [28]. (d) The CHSs offer various advantages in structures exposed to fluid currents because these sections give minimum resistance to water and air wind–wave loadings, as shown in Figure 1. (e) In CHSs, the section strength can be improved by increasing the wall thickness or filling the section with concrete, without changing the outward measurements. (f) The internal void of a CHS provides a combination of high strength with other features, such as fire protection, ventilation, and proper heat transfer ability, which in turn facilitate the placing of electrical wires inside the beams and columns [29]. The effect of the presence of openings in the web of steel beams has been studied experimentally and analytically by Morkhade et al. [30,31,32,33]. The study shows the effect of the opening on the ultimate load behavior and various failure modes.

It has been observed in the literature that there are many existing experimental works to predict the behavior of steel tubes [34,35,36,37]. However, there is no available experimental work to evaluate the effect of diameters, section thickness, and beam span with and without openings on the structural behavior of circular hollow steel tube sections, as well as their mechanical performance, which is important to investigate under different loadings; this information can be used to be a reference for future studies and provide extensive data for steel tube studies.

## 2. Materials and Methods

This section describes the used materials, specimen preparation, testing equipment, and methods used to determine the properties during the experimental research.

### 2.1. Material Properties

Steel is the only material used during the experiment. Steel comprises iron and other elements, such as carbon, manganese, phosphorus, sulfur, nickel, chromium, and more. Variations in steel compositions are responsible for the great variety of steel grades and steel properties. Iron is the basic component of steel. The tensile specimens were prepared as per the ASTM (A370) specifications [38], as depicted in Figure 2. Prepared specimens were tested to evaluate the yield stress (F_y_) and the ultimate stress (F_u_) using a tensile testing machine, as tabulated in Table 1 and Figure 3. The specimens were loaded in the longitudinal direction in steps until the specimen’s failure occurred.

### 2.2. The Experimental Plan

The experimental plan comprised the evaluation of ten beam specimens from the pre-failure stage to the post-failure stage with various circular section sizes and diameter-to-thickness ratios ranging from 16.93 to 73.

These specimens were distributed as follows.

#### 2.2.1. The Reference Specimen

One specimen was chosen as a reference for comparing the results. This specimen had a span of 1500 mm, a diameter of 101.6 mm, and a thickness of 3 mm.

#### 2.2.2. The Modified Specimens

The modified specimens were divided into four different groups as follows.

##### The First Group

With this group, we aimed to recognize the effect of changing the thickness of the circular hollow section on the mechanical behavior. The section thickness is the most important characteristic in this group. We used two specimens whose thickness differed, while the span and diameter were constant, as shown in Table 2.

The selected variable is essential since the circular hollow steel section’s buckling behavior and bending capacity depend mainly on the section’s diameter-to-thickness ratio.

##### The Second Group

With this group, we aimed to recognize the effect of the presence of openings on the investigated mechanical properties. The presence of square apertures is the major variable in this category. We used three specimens whose opening numbers and locations in the specimen were different, while all other parameters were constant, as shown in Table 2.

The selected variable is essential since the functional requirements are a significant aspect of the design considerations. The use of steel beams with apertures has various practical applications, such as ventilation, heating systems, and pipelines, which have broad usage.

These specimens are as follows:

The first specimen, BT1, is the reference specimen used without openings;

The second specimen, BT4, is a specimen with one opening located at the center of the specimen;

The third specimen, BT5, is a specimen with two openings located at the two loading points of the specimen;

The fourth specimen, BT6, is a specimen with three openings located at the center and two loading points of the specimen.

The openings used in each of these specimens were identical in their shape and dimensions, which were square shapes with lengths equal to 50 mm.

##### The Third Group

With this group, we aimed to recognize the effect of changing the span of the circular hollow specimens on their investigated properties. The span of the beam specimen is the crucial variable in this group. We used two specimens whose span was different, while all other parameters were constant, as shown in Table 2.

##### The Fourth Group

With this group, we aimed to recognize the effect of changing the outside diameter of the circular hollow specimen on the investigated properties. The primary variable of this group is the outside diameter of the specimen (Table 2). We used two specimens whose outside diameter was different, while all other parameters were constant.

The selected variable is essential since the buckling behavior and bending capacity for circular hollow sections depend mainly on the diameter-to-thickness ratio of the circular sections.

### 2.3. Test Procedure and Equipment Used

Once hollow cylindrical tubes are bent, a deflection in their cross-section develops, resulting in ovalization. The ovalization phenomenon will develop with the increase in bending moment, resulting in a steady decrease in the specimens’ bending stiffness. When the degree of the ovalization exceeds the critical value, the circular hollow specimens will be susceptible to local buckling [5]. As a result, appropriate conditions must be created at the specimen’s loading locations to avoid early ovalization and restrain the circular portions; hence, the cross-section will be forced to remain circular during the loading. In this experimental work, four circular rings were employed to place the specimen at the supports and two loading positions to accomplish this. These circular rings had an internal diameter equal to the external diameter of the specimen. Each ring comprised two semi-circular pieces, whose width and thickness equaled 35 mm, connected by bolts, as shown in Figure 4.

The rings acted as a stiffener for the specimens, and their functions were as follows:To ensure that the applied load is distributed axially to avoid stress concentration at a single place;To support the vertical load;Due to the high rigidity and stiffness of these rings, radial displacement at loading locations is limited;The compression section of the specimen will be stiffened and supported adequately, preventing local buckling at the stress locations;To prevent the specimens’ sudden failure when they reach the peak load.

A Hydraulic Universal Testing Machine (MFL system) is a machine having a maxi-mum range capacity of 3000 kN, used for testing slab specimens. Figure 5 displays this machine. A four-point flexural test was employed to investigate the specimens’ properties. Each specimen was located between the supports, as shown in Figure 5b. A hydraulic jack connected to a load cell and attached to the beam was used to apply a vertical load on the specimen’s center. Through the beam, the applied force was equally distributed to the specimen at two loading positions. The loading points were located at equal distances from the center of the specimen: a cylindrical roller was placed between the spreader beam and the circular rings surrounding the specimen.

### 2.4. The Measurement Instruments

Dial gauges

Five dial gauges were used for each specimen test and distributed as follows.

One of the dial gauges was placed in the center of the specimen, and two gauges were placed at the loading points to measure the vertical deflection at these points, as shown in Figure 6.

One of the remaining dial gauges was placed in the front face (front side) of the specimen and parallel to its center, while the second dial gauge was placed in the back face (back side) in order to measure the ovalization deformation of the cross-section at mid-span, as shown in Figure 7.

Data logger

A data logger was used to record all data measured using the strain gauges in the specimen, providing the dependence of strain on the time. The device is shown in Figure 8.

Strain gauges

Twelve strain gauges whose length was equal to 10 mm were used for each specimen to measure its strain, and a TML strain gauge was used in this study, as shown in Figure 9. The use of the mentioned strain gauges was justified by the high degree of stability, accuracy, and ease of use.

The PFL-10-11-3L strain gauge type was used in the present study, with the following properties: a single element had a pre-attached vinyl lead wire with 3 m. The gauge had a length of 10 mm, a width of 2.3 mm, a gauge factor of 2.12 ± 1%, and resistance of 119.5 ± 0.5 Ω.

Four strain gauges were used in each location, as shown in Figure 10. These strain gauges were identically distributed at the center of the specimen and two loading points. The strain gauges were distributed at the top, the bottom part of the cross-section, the front face, and the back face to measure the specimens’ strain at these locations.

For the installation of the strain gauges on the specimen, the following procedure was adopted.

Preparation of the surface: The bonding area was cleaned of rust, grease, and paint and uniformly finished using abrasive paper #120 to #180.

Fine cleaning: The bonding area was cleaned by using a cloth soaked in acetone.

Bonding the strain gauge: The adhesive CN series was used to bond the strain gauges, cured at room temperature with the required time of approximately 40–120 sec.

Coating the strain gauge: The SB tap was used to coat the strain gauges with 5–10 mm length. It offers excellent water and moisture resistance characteristics (Figure 11).

## 3. Results and Discussion

### 3.1. General Behavior of the Investigated Specimens under Loading

When subjected to a bending moment, the behavior of circular hollow steel is characterized by three different stages. These stages are:-The elastic stage;-The ovalization stage;-The collapse stage.

Firstly, the material shows a linear elastic response. After this, the cross-section gradually changes from a circular to an oval shape. This process continues to a limit point, which is the maximum bending capacity of the specimen. The specimen ovalization becomes localized deformation at this point, which is distinguished by the formation of a kink or fold in the specimen’s compression section.

According to the general observations of the investigated specimens, the following general behavior could be concluded.

According to the observations, increasing the specimen thickness improves the strength capability and the specimen’s ability to absorb and dissipate energy, which delays the specimen’s failure; at the same time, it prevents a rapid drop in the load-carrying capacity.

When decreasing the specimen’s span, the specimen’s strength capacity was improved, but its ability to absorb energy and ductility decreased.

Moreover, increasing the specimen’s diameter enhances the strength capacity significantly. However, it negatively influences the specimen’s ability to absorb and dissipate energy, causing a sudden drop in the load-carrying capacity at the ultimate loading points.

The presence of openings did not influence the general behavior of the investigated specimens; however, a fraction of the strength was lost. The strength capacity of the specimen remains unaffected by the number of openings in the pure bending region.

### 3.2. Results of the First Investigated Group

The strength characteristics of the first group of specimens are shown in Figure 1.

#### 3.2.1. Load–Deflection Curve

As illustrated in Figure 2, the behavior of the tested specimens under load was dictated mostly by the specimen’s actual thickness. According to the general stages of circular hollow steel, the following behaviors were examined.

1.The elastic stage

All the tested specimens had linear load–deflection relationships at the initial loading end at the yield point. The yield points of the tested samples BT1, BT2, and BT3 were equal to 45, 30, and 85 kN, respectively, as shown in Figure 2.

The yield load value of specimen BT2 was lower by 33.33% compared with the reference specimen, BT1. In comparison, the yield load of specimen BT3 was greater than that of the reference specimen by 88.89%, corresponding to the increase in the specimen’s thickness, as shown in Figure 1.

2.The ovalization stage

This stage began at the yield load, which represents the first point of the transformation curve to a flat line, with a slight inclination due to the high increase in the specimen’s deflection.

The ovalization stage of the BT1 and BT3 specimens was broader than that of the BT2 specimen, as seen in Figure 2. It is supported by the increased thickness of the BT1 and BT3 specimens, which enhanced the strain-hardening capacity of these specimens, resulting in significant stress redistribution until they reached the ultimate load. Therefore, the ultimate load for the BT1, BT2, and BT3 specimens increased directly with the specimen’s thickness. The increase in the thickness by 200% leads to a corresponding increase in the specimen’s ultimate load by 182%. At the same time, a decrease in the specimen’s thickness by 150% leads to a corresponding decrease in the specimen’s ultimate load by 61%.

3.The failure stage

This stage begins with the ultimate load and ends with the specimens failing. Under pure bending, the reference specimen BT1 began to collapse following smooth kink development in the compressive zone (top surface for BT1 specimen). The local buckling of the specimen at the top surface is caused by the load increment after kink formation, which increases the deformation and causes gradual kink growth. Local buckling leads to the twofold formation, and, for the BT1 specimen, the fold at the top surface exposed to compressive stresses and the fold at the bottom surface are subjected tensile stress. Finally, as indicated in Figure 12, a widespread buckling failure mode occurs.

In contrast, the collapse of specimen BT2 was initiated after the development of wave buckling on the upper region (compressive zone) at the specimen mid-span, as shown in Figure 13. The amplitude and wavelength gradually increased at the top surface on increasing the load, followed by a high increase in the specimen’s deflection. Stress redistribution occurred after the ultimate load point (13 kN), which increased the load-carrying capacity of the specimen due to the strain hardening, which lasted until the fracture load.

After the ultimate load, the failure stage in the BT3 specimen began without any buckling and was accompanied by an increase in the specimen’s deformation. The applied loads on this specimen caused high compressive stresses in the top surface, while the bottom surface was subjected to tensile stresses. This led to a high curvature in the specimen’s central region, as shown in Figure 14.

It could be observed that increasing the specimen’s thickness enhanced the load-carrying ability of the specimen and transformed the cross-section’s structure from non-compact in specimen BT2 to compact in specimen BT1 and plastic in specimen BT3. The failure mechanism changed from increased curvature without buckling in the BT3 specimen to wave buckling in the BT2 specimen and smooth kink development in the BT1 specimen as the specimen thickness changed.

#### 3.2.2. Ductility

Ductility is an important mechanical property of steel specimens that measures the degree of the specimen’s plastic deformation. According to Sabouri and Ghol [1], ductility intensity can be calculated by dividing the ultimate deflection by the yield deflection at the mid-span of a specimen. Figure 1 shows that specimen BT2 had low ductility (up to 2.47) due to the lower ultimate deflection, which caused a sudden sharp drop in the load-carrying capacity after the application of the ultimate load, as shown in Figure 2. Specimen BT1 had higher ductility (up to 7.03), due to the much higher deformation at the ultimate load as compared to yield deformation at yield load, and this caused a gradual drop in the specimen’s load capacity. Much higher ductility (up to 11.11) was observed in specimen BT3, which caused a gradual and flat drop in the ultimate load-carrying capacity of the specimen.

The above observations indicate a proportional relationship between the ductility and the specimen thickness; therefore, increasing the specimen thickness caused an increase in the ductility and ultimately caused a gradual drop in the load-carrying capacity up to the fracture load.

#### 3.2.3. Stiffness

Stiffness measures the specimen’s resistance to deformation and relates to how the specimen bends under the applied load. The specimen stiffness was calculated at 45% of the ultimate load. Figure 1 shows the stiffness values for the first group of specimens. It can be concluded that the increase in the BT3 specimen’s thickness led to a corresponding increase in its stiffness by 22.66% compared with the reference specimen, BT1. In contrast, the thickness reduction in specimen BT2 caused a significant decrease in its stiffness by 49.21% compared with the reference specimen, BT1, and led to a decrease in the specimen’s resistance to deformation. Meanwhile, the reference specimen BT1 had a stiffness value equal to 8.21 kN/mm.

According to the above, it appears that the specimen’s stiffness is directly proportional to its thickness.

#### 3.2.4. Deflection Profile

Figure 3 shows a comparison of the deflection profiles at yield and ultimate load for the first group of specimens, BT1, BT2, and BT3. In comparison to specimens BT1 and BT2, specimen BT3 underwent higher deformation at the mid-span under the influence of yield load. The greatest deflection of specimen BT3 under ultimate load was 31.84 percent greater than that of the reference specimen, BT1. The peak deflection at yield load for both the BT2 and BT3 specimens was similar and 21.14 percent higher than the comparable deflection of reference specimen BT1.

At the ultimate load, these specimens behaved similarly, but the observed deflection values differed. Focusing on Figure 3, it can be noted that specimen BT3 shows a much higher increase in its deflection, with a peak deflection of 108.06 mm, which is almost twice the peak deflection of reference specimen BT1. Specimen BT2 shows a lower response at ultimate load, and its peak deflection value is 57.40% less than that of reference specimen BT1.

According to the findings, increasing the specimen’s thickness increases its ability to absorb and dissipate energy, as well as its resistance to failure when subjected to ultimate stress.

#### 3.2.5. Cross-Section Ovalization Behavior

Figure 4, Figure 5 and Figure 6 illustrate the cross-section behavior of specimens BT1, BT2, and BT3 during the loading procedure. Figure 4 shows that the cross-section of the specimen BT1 did not show any response in the elastic zone, but when the yield load of 45 kN was applied, the back side of the cross-section began to deform, accompanied by the curvature of the specimen. As shown in Figure 5, the cross-section of specimen BT2 reveals its response from the start of loading. The exposure of the front side to tensile deformation and the back side to compressive deformation gradually changed the cross-section; this occurrence is known as specimen ovalization.

The tensile deformation on the front side and compressive deformation on the back side of specimen BT3 triggered its response to applied loads after 10 kN load application, as illustrated in Figure 6. With increasing loads applied on this specimen, these deformations continued to increase until the yield load of 85 kN was reached. After the yield load, the strain hardening caused a redistribution of stresses, resulting in tensile deformation in specimen BT1. The front side of specimen BT3 was also exposed to compressive deformation at the same time.

Increasing the loads on specimens BT1, BT2, and BT3 to an ultimate load of 54.8, 33.5, and 99.6 kN, respectively, resulted in an increase in cross-section deformation and transformation to an oval shape, as well as a decrease in bending stiffness. As demonstrated in Figure 15, the ovalization phenomenon for these specimens was transformed to localized deformation by the creation of a kink and wave buckling in the compression zone (top surface) of specimens BT1 and BT2, respectively. Both sides of the cross-section for specimens BT1 and BT2 responded identically when exposed to tension deformation. The top and bottom surfaces’ curvature increased as the BT3 cross-section of the specimen gradually changed. The negative stiffness then took control of the specimens, rendering them unstable and eventually causing structural collapse.

Figure 7 shows how the change in D/t ratio affects the cross-section diameter at the mid-span for specimens BT1, BT2, and BT3. When compared to the initial diameter, the diameter of the reference specimen BT1 expanded by 8.27 percent at the maximal load. Due to wave buckling on the top half, which resulted in reduced cross-section ovalization, specimen BT2 displayed a very small increase in diameter, equal to 0.6 percent, when compared to the starting diameter. When compared to the initial diameter, the cross-section diameter of the BT3 specimen was reduced by 1.56 percent.

The above result shows that the ovalization phenomenon for specimens is affected by the change in the diameter-to-thickness ratio. The ovalization of the cross-section is negligible when the D/t ratio of the specimens is less than 16.93 or greater than 50.8, according to the obtained results, because the specimens with a D/t ratio greater than 50.8 were controlled by local buckling failure, which resulted in the non-appearance of the cross-section ovalization. However, the specimen’s diameter increases when the D/t ratio ranges between 16 and 50, leading to the appearance of ovalization. The Results of the Present Group Are in Line with a Study Conducted by He, P. and Pavlovic, M. [36].

### 3.3. Results of the Second Investigated Group

The primary variable of this group is the presence, number, and location of square openings in the specimen. Figure 8 shows the strength characteristics of the second group of specimens.

#### 3.3.1. Load–Deflection Curve

From Figure 9, it can be noted that specimens BT4, BT5, and BT6 underwent the same stages as the reference specimen BT1 when loaded gradually. These stages are as follows.

1.Elastic stage

The elastic stage began with the initial loading and ended with the yield load for specimens BT4, BT5, and BT6. These specimens’ yield points were 40 kN, 41 kN, and 40 kN, respectively, and they were characterized by a linear connection between the applied stress and specimen deflection.

In this stage, all the specimens of the second group had very similar elastic behavior, as a similar variation was observed in the yield load with the deflection values. From Figure 9, it can be noted that the presence of openings in the specimens did not significantly influence the elastic stage for these specimens, because these openings were located at the neutral axis of the cross-section, and thus they had a negligible effect within this stage.

2.The ovalization stage

This stage marks the beginning of the specimen’s plastic behavior. The applied loads increased slightly compared to those with a higher increase in the specimen deflection until the ultimate load was obtained. The plastic deformation caused by combined moment and shear forces at the openings determined the specimens’ strength capability.

The moment capacity of these specimens was reduced significantly, and the ultimate load values of specimens BT4, BT5, and BT6 were reduced by 17.88%, 19.71%, and 14.23%, respectively, as compared to the reference specimen BT1.

The presence of the openings affected the ovalization stage by reducing the strain-hardening capacity, which led to a significant reduction in the stress redistribution of the specimens compared with the reference specimen BT1.

3.The collapse stage

Tensile stresses cause the bottom half of the specimen below the holes to yield. The upper half of the specimen, above the aperture, on the other hand, was subjected to yield and buckling, resulting in a large relative deflection between the opening ends. The kink formation caused by high plastification on the upper surface of the apertures due to high compressive stress caused specimen BT4 to collapse. This kink matured into local buckling, which spread from the top surface of the opening to the top surface of the specimen, forming two folds. The two folds were located on the top and bottom surfaces, resulting in the general buckling failure mechanism illustrated in Figure 16.

The fold formation in the specimen’s top surface at the opening in the left loading point initiated the failure of specimen BT5. As illustrated in Figure 17, continuing to load the specimen after the ultimate load increased the amplitude and wavelength of the fold, without causing transmission to other sections of the span. Finally, this fold induced local buckling failure in the specimen’s compressive zone.

On applying the ultimate load on the specimen BT6, the top surface of the central opening experienced significant distortion due to yielding, which was followed by buckling failure. As illustrated in Figure 18, the continued loading process on this specimen induced local buckling that spread from the center aperture to the top surface of the specimen, leading to the creation of two folds that increased the specimen’s curvature and eventually led to general buckling failure.

It can be noted that the opening number does not influence the behavior in the pure bending region, and, thus, it has a negligible effect on the structural behavior.

#### 3.3.2. Ductility

The ductility values of the second group of specimens, BT1, BT4, BT5, and BT6, are shown in Figure 8. When compared to the reference specimen BT1, the existence of apertures in the specimens lowered their ductility by 72.40%, 67.71%, and 60.88%, respectively. As a result, at the maximum load, these specimens’ load-carrying capacity declined considerably compared to the gradual reduction observed in the reference specimen BT1, as seen in Figure 9.

#### 3.3.3. Stiffness

Figure 8 shows the stiffness values for the second group of specimens. The results show that the presence of openings in specimens BT4, BT5, and BT6 reduced their stiffness by 39.83%, 20.22%, and 41.90%, respectively, compared with the reference specimen BT1. Thus, it can be noted that the central opening has the most significant effect on the specimen’s stiffness compared to other openings.

#### 3.3.4. Deflection Profile

A comparison of the deflection profiles is shown in Figure 10. Specimens BT5 and BT6 exhibited similar loading behavior at yield load. These specimens’ maximum deflections were 10.20 and 9.51 mm, respectively, at the left loading point. The highest deviation of specimen BT4 was 9.07 mm at the specimen mid-span, which was approximately equivalent to the mid-span deflection of specimens BT5 and BT6. Specimens BT1, BT4, and BT6 behaved similarly under ultimate load but had different deflection values. In comparison to the similar circumstances of the reference specimen BT1, the maximum deflection of specimens BT4 and BT6 at the ultimate load was 34 percent and 47 percent, respectively.

When compared to the similar deflection conditions of the reference specimen BT1, the deflection at the mid-span of the BT5 specimen reached up to 40% at the highest load. The highest deflection value reached 51.88 mm in the reference specimen BT1, which had the most responsiveness at the mid-span compared to the other specimens.

According to our observations, the presence of openings reduced the ability of specimens BT4, BT5, and BT6 to absorb and dissipate energy, lowering their resistance to fracture and ultimately accelerating the specimen’s failure, in contrast to the reference specimen BT1, which was characterized by its ability to resist failure.

#### 3.3.5. Cross-Section Ovalization Behavior

The cross-section behavior of specimens BT4, BT5, and BT6 is shown in Figure 11, Figure 12 and Figure 13, respectively, under loading conditions. During the initial loading stage, the cross-section started to ovalize as a result of tension deformation on the front side and compression deformation on the back side in specimen BT4, as shown in Figure 11, and compression deformation on the front side and tension deformation on the back side in specimen BT5, as shown in Figure 12. When the applied load was increased, both sides of specimen BT6 displayed compression deformation; however, the front side showed greater reactivity than the rear, as illustrated in Figure 13. Increasing the applied loads on specimens BT4, BT5, and BT6 induced gradual ovalization of the cross-section and a significant drop in bending stiffness; this situation persisted until the ultimate load reached 45, 44, and 47 kN, respectively. Due to the high plastification that occurred in the top surface of the opening, leading to ovalization transformation into localization deformation characterized by the formation of the kink in this surface accompanied by the exposure of both sides to tension deformation, the front side of specimen BT4 began to compress at the ultimate load.

However, at ultimate load, a significant increase in its response was observed, followed by the formation of a fold in the top surface at the left loading point within the pure bending zone. As demonstrated in Figure 19, this fold was linked to the localization phenomenon and produced structural collapse. When specimen BT6 achieved its ultimate load, tension distortion was seen on the back side, while compression deformation on the front side continued to rise according to the applied stresses. Furthermore, due to the negative stiffness, kinks formed on the upper surface of the central aperture, causing the specimen’s structural collapse.

Figure 14 shows how the D/t ratio affected the diameters of the second group of specimens at the ultimate load. The ovalization phenomenon caused the cross-sections of specimens BT4, BT5, and BT6 to show a decrease of 0.94%, 1.23%, and 4%, respectively, compared to the earlier diameter, due to the presence of openings in the front side of the specimen’s cross-section, which buckled and caused deformation and a lack of clarity.

The results of the present group are in line with a study conducted by Zeinoddini, M., et al. [39].

### 3.4. Results of the Third Investigated Group

Figure 15 shows the strength characteristics of the third group of specimens.

#### 3.4.1. Load–Deflection Curve

Figure 16 shows the load–deflection curve of the third group of specimens, BT1, BT7, and BT8. The behavior of the specimens under loading is explained as follows.

1.The elastic stage

For specimens BT7 and BT8, the elastic stage started from the initial loading to yield loads equal to 38 and 90 kN. It is characterized by high load change with little change in the specimen’s deflection. The change in the specimen span significantly affected the specimen’s elastic stage; therefore, the yield load value of specimen BT8 increased by 100% compared with the reference specimen BT1, as shown in Figure 16, while, for specimen BT7, its value decreased by 15.56%. This was due to the span reduction of specimen BT8, which caused an increase in the specimen’s load capacity and increased the specimen’s linear elastic stage.

2.The ovalization stage

This stage is characterized by high deflection change compared to the applied loads exposed to the specimens. Specimen BT7 possessed an ultimate load equal to 53 kN, which is considered low compared with specimen BT8, which was characterized by a high ultimate load reaching 104 kN. This is justified by the length being inversely proportional to the load-carrying capacity of the specimen.

The failure stage

The deflection and curvature of the top and bottom surfaces of the BT7 specimen increased dramatically during the failure stage. It was found that at a load of 35 kN, the top surface in the center region of the specimen was subjected to considerable compression stress, resulting in the formation of a smooth kink and a significant rise in deflection. It led to the formation of two folds, one on the top surface and the other on the bottom surface, and, finally, as illustrated in Figure 20, worldwide buckling failure. The failure stage for the BT8 specimen started with the introduction of a fold in the top half of the specimen, close to the right loading ring. The maximal load was accompanied by a small increment in specimen deflection when applied. The fold developed as a result of the continuous specimen loading procedure, increasing its amplitude without impacting its length. Finally, as illustrated in Figure 21, it resulted in local buckling failure in the compression section.

According to our observations, increasing the specimen length changed the type of specimen failure from local buckling in specimen BT8 to general buckling in specimens BT1 and BT7 through kink formation.

#### 3.4.2. Ductility

Figure 15 shows the ductility values of the third group of specimens. Due to the low value of deflection at the ultimate stress, the BT8 specimen had poor ductility compared to the standard BT1 specimen, which was equal to 3.50. The load-carrying capacity of this specimen dropped sharply and abruptly as a result of this. While the deflection of the BT7 specimen was considerable at both yield and ultimate load, this resulted in moderate specimen ductility and an equal value of 4.42.

#### 3.4.3. Stiffness

Figure 15 illustrates the stiffness values of the third group of specimens. The rigidity of the specimens was lowered when the length of the specimens was increased. When compared to the reference BT1 specimen, the stiffness of the BT7 specimen was reduced by 58.34%. Because specimen stiffness is inversely related to specimen length, the final deflection of specimen BT8 was equivalent to 25.80 mm, resulting in a 58.59% increase in specimen stiffness compared to the reference specimen BT1.

#### 3.4.4. Deflection Profile

Under yield and ultimate load conditions, Figure 17 compares the deflection profiles of the third group, BT1, BT7, and BT8. The results demonstrate that, under the impact of yield load, specimen BT7 had higher deflection than specimens BT1 and BT8, with a maximum deflection value of 69.38%, higher than the corresponding deflection of reference specimen BT1. The maximum deflection of specimen BT8 under this load, on the other hand, is comparable to the deflection of the reference specimen, which was 7.38 mm.

Specimens BT1, BT7, and BT8 behaved similarly under the ultimate load condition; however, their deflection values varied. The BT7 specimen showed a large rise in deflection values, with a maximum value of 55.31 mm, but the BT8 specimen exhibited a lower response at the same load, with a maximum deflection value of 50.27%, less than the reference specimen.

The results demonstrate that lengthening the specimens enhances their ability to absorb and dissipate energy.

#### 3.4.5. Cross-Section Ovalization Behavior

When the first loading occurred, the cross-sections of specimens BT7 and BT8 began to shift. The front side of the BT7 specimen first displayed a minor compression deformation response. The back side of this specimen developed compression deformation with higher applied loads when subjected to modest tension deformation, as demonstrated in Figure 18.

The BT8 specimen exhibited compression deformation on both sides, which led to the ovalization of the specimen due to a gradual change in cross-section diameter, as illustrated in Figure 19. The decrease in bending stiffness caused by the cross-section change ended in the localized ovalization phenomenon. Kink formation in the top portion of the BT7 specimen and fold formation alongside the right ring in the compression area of the BT8 specimen identified the location. Following this, the specimens failed due to progressive localization expansion, as illustrated in Figure 22.

As shown in Figure 20, specimens BT7 and BT8 displayed a reduction in cross-section diameter of 2.45% and 1.33%, respectively, at the ultimate load, compared to the initial diameter. As a result, it appears that changing the specimen length did not play a role in cross-section ovalization.

The results of the present group are in line with a study conducted by Zhao, O. et al. [40].

### 3.5. Results of the Fourth Investigated Group

Figure 21 shows the strength characteristics of the fourth group of specimens.

#### 3.5.1. Load–Deflection Curve

Figure 22 shows the behavior of the fourth group of specimens, BT1, BT9, and BT10, which can be explained as follows.

1.The elastic stage

For the BT9 and BT10 specimens, this stage was characterized by a linear relationship between the applied loads and the specimen deflection, which was prolonged to a yield load equal to 160 and 17.5 kN, respectively. The obtained results show that the increase in the specimen’s diameter increased the values of the yield load significantly and led to an increase in the elastic stage.

2.The ovalization stage

For specimen BT10, this stage was characterized by the curve’s transformation from a straight line to a flat line due to the high increase in the specimen’s deflection, compared to the slight increase in the specimen loads. In contrast, the ovalization stage for the BT9 specimen was characterized by behavior in which the specimen showed a small increment in deflection with a higher increment in the applied loads. This stage for specimens BT9 and BT10 continued to an ultimate load equal to 185 and 23.7 kN, respectively.

Further, the ultimate strength capacity showed a proportional relationship with the diameter of the specimen; hence, BT1 and BT9, specimens with high diametric values, exhibited higher strength compared to specimen BT10, with a smaller diameter.

3.The failure stage

Fold (outwards bulge) formation was observed in the compression part of the specimen cross-section adjacent to the right loading point ring at the ultimate load. The ongoing loading process caused the formation of this fold, i.e., outward bulge, and increased the amplitude of the fold. At a higher load, 100 kN, after the ultimate load condition, the high strain-hardening ability of the specimen resulted in the redistribution of the specimen’s stresses, which enhanced the capacity of the specimen from 107 to 130 kN. However, at a later stage, another fold formed in the specimen’s cross-section adjacent to the left loading point ring, which caused a reduction in the load-carrying capacity. As illustrated in Figure 23, both of these folds continued to expand with applied loads up to a certain point. Finally, the compression section and the area outside the center span experienced local buckling failure. The high diameter, which increases the specimen cross-section resistance beside the rings, supports the formation of these folds beyond the specimen’s central span area.

The collapse of the BT10 specimen began with high deflection at the ultimate load, and it increased with the ongoing loading process and led to the development of a strong kink in the compression part of the central span region. Finally, it led to the occurrence of two folds, one on the top surface and the other on the bottom surface, accompanied by a rapid drop in the load-carrying capacity, and thus the global buckling failure occurred, as shown in Figure 24.

Results show that varying the specimen’s diameter from 219 mm to 101.6 and 76.2 mm changed the cross-section’s structure from a non-compact section in the BT9 specimen to a compact section in the BT1 and BT10 specimens. Consequently, the mode of failure also changed to general buckling from local failure.

#### 3.5.2. Ductility

Increasing the specimens’ diameter affected their ductility, as shown in Figure 21. The plot shows that the ductility of the BT9 specimen was very small and had a value of 1.64 at the ultimate load, which caused a swift and unexpected descent in the load-carrying capacity of the specimen. However, the BT10 specimen showed higher deflection values at yield as well as ultimate load, which in turn improved the ductility value to 4.80 and caused a gradual drop in the load–deflection curve after the ultimate load.

#### 3.5.3. Stiffness

Figure 21 shows the stiffness values for the fourth group of specimens, and the given results show that the stiffness is directly proportional to the specimen diameter. The BT9 specimen had high stiffness up to 26.40 kN/mm. In contrast, the BT10 specimen possessed low stiffness of 1.92 kN/mm.

#### 3.5.4. Deflection Profile

Figure 23 shows the deflection profile for the fourth group, BT1, BT9, and BT10, under the load condition at the yield point and ultimate point. Under the loading condition at yield load, the BT10 sample displayed a high response, with a maximum deflection value at mid-span increased by 64.50% from the corresponding deflection of the reference specimen, while the BT9 specimen showed a deflection value equal to 8.65 mm at the same conditions. At the ultimate load, the BT1 and BT10 specimens behaved similarly. Specimen BT10 exhibited a higher response and increased the maximum deflection at the mid-point of the specimen by 12.28%, compared with the maximum deflection of specimen BT1. Meanwhile, specimen BT9 revealed a weaker response at this load and reduced the deflection magnitude at the mid-point by 72.73% from the corresponding deflection of reference specimen BT1.

From the above observations, it can be noticed that the diameter of the specimen is inversely related to its ability to absorb and dissipate energy and its resistance to fracture, i.e., the higher the diameter, the less it will absorb and dissipate the energy. Further, higher values of specimen diameter eventually lead to the failure of these specimens.

#### 3.5.5. Cross-Section Ovalization Behavior

A very smooth change occurred in the cross-section of the BT9 specimen at the initial loading, with a minimal effect through the tension nature of deformation in the front side while the compression at the back side took place, as depicted in Figure 24. Meanwhile, both sides of specimen BT10 began responding through tension deformation, as shown in Figure 25. After the yielding point, the stresses redistributed themselves and resulted in different types of deformation at the front and back sides of the BT10 specimen, i.e., tension and compression deformation at the back side and front side, respectively, which continued with the increment in the applied loads. At yield loading, a reduction in the front side deformation was observed in the BT9 specimen.

This alteration in the cross-section resulted in a reduction in the bending stiffness of the specimens. Therefore, the ovalization transformed into the localization phenomenon by the fold composition in the compression part of the BT9 specimen at the right loading point, accompanied by the transformation of the tension deformation in the front side into compression deformation. However, the BT10 specimen showed localization with kink formation in the compression part, which eventually led to failure, as demonstrated in the load–displacement plot (Figure 26 and Figure 25).

The cross-section diameter of specimen BT10 was increased by 4.46% compared with the initial diameter, shown in Figure 26. At the same time, the cross-section of the BT9 specimen was reduced by 0.49% compared with the initial diameter due to the specimen’s instability. The results of the present group are in line with studies conducted by Kozich, M., & Wald, F. [41] and Zhou, W. et al. [42].

### 3.6. Mid-Span Strain Distribution and Progression of the Neutral Axis Depth

This section presents the strain gauge readings for the test specimens at the top and bottom surfaces in the mid-span location at different load levels, used to determine the neutral axis location (N.A.) for the test specimens. Therefore, curves were drawn that illustrated the change in the N.A. location, which is c/d (ratio of the depth of compression zone to the test specimen diameter), with the change in applied load. The strain readings at the front and back of the specimen cross-sections were slight.

#### 3.6.1. The First Group

The strain distribution at the top and bottom surfaces in the mid-span locations for the BT1, BT2, and BT3 specimens is shown in Figure 27, Figure 28 and Figure 29.

The obtained results show that the gradual loading of the BT1, BT2, and BT3 specimens led to equal strain values in the compression strain at the top surface and the tension strain at the bottom surface. Therefore, the N.A. location remained constant at the initial loading up to loads equal to 36, 20, and 91 kN, respectively, as shown in Figure 27. After these loads, the N.A. location moved to the top and caused a reduction in the compression zone depth due to the high increase in the tension strain compared with the compression strain due to the presence of rings that supported and tightened the specimen compression part. Within this stage, the relationship between the applied load and the resulting strain was linear.

Thus, it could be concluded that the specimen thickness affects the compression and tension strain and the N.A. location; therefore, increasing the specimen thickness increased the compatibility between the tension and compression strain and led to the steadiness of the N.A. location in the BT3 specimen, which continued beyond the yield load. Meanwhile, the location of the BT1 and BT2 specimens began to change at 88.89% and 83.33% from the yield load, respectively.

The stress redistribution occurred in specimens BT1 and BT3 due to these specimens’ good strain-hardening capacity, which caused an increase in the compression strain values that was more significant than the tensile strain. In turn, it led to the descent of the N.A. location to the bottom of the specimen center, and thus the depth of the specimen compression zone increased.

This case remained during the ongoing loading of specimens BT1 and BT3. When specimen BT1 reached a load equal to 54 kN, it was found that the value of compression strain became equal to 35,094 με, and this indicates that the top surface reached the yield stress, while the tension strain value was 47.46% lower than the compression strain. However, after this load, the compression strain value decreased with increasing specimen deformation due to the buckling failure in the top surface of the specimen, while the tension strain continued to increase with increasing applied loads. When specimen BT3 reached the ultimate load, the strain at the bottom surface increased significantly with increasing specimen deflection and caused the N.A. location to move upward, accompanied by a decrease in the depth of the specimen compression zone.

The stress redistribution in the BT2 specimen did not increase the compression strain, and when the specimen reached the ultimate load, it was observed that the c/d ratio was 0.13 because of the high increase in tension strain compared with a slight increase in compression strain. At this load, the measured value of compression strain was equal to 2077 με, which represents 7.29% of the yield strain. Thus, the elastic local buckling deformation appeared in the top surface of the specimen, accompanied by a large reduction in compression strain.

The specimen thickness has an obvious effect on the strength development and the type of failure in the specimen compression part; therefore, increasing the specimen thickness led to the occurrence of yielding without the appearance of any buckling, as in the BT3 specimen, and the occurrence of yielding followed by buckling in the BT1 specimen, while decreasing the thickness of specimen BT2 led to the occurrence of buckling without yielding.

#### 3.6.2. The Second Group

The strain distribution at the top and bottom surface in the mid-span locations for the BT4, BT5, and BT6 specimens is shown in Figure 30, Figure 31, Figure 32 and Figure 33, respectively. Results show that the gradual loading of the BT4, BT5, and BT6 specimens led to a linear relationship between the applied load and resulting strain up to yield load equal to 40, 41, and 40 kN, respectively. At these loads, the strain values at the top and bottom surface were almost equal, caused the N.A. location remaining constant up to a load equal 35, 38, and 35 kN for specimens BT4, BT5, and BT6, respectively, as shown in Figure 22. The N.A. location of the specimens transferred to the top, leading to a decrease in the compression zone at the yield load. Compared to the tensile strain, the compression strain of these specimens was reduced by 10.48%, 6.66%, and 42.81%, respectively. After the yield load, the stress redistribution for the BT4 and BT6 specimens increased the compression strain values, increasing the depth of the specimen’s compression zone. Meanwhile, the stress redistribution for the BT5 specimen did not lead to the increased depth of the compression zone due to the local buckling at the left loading point, which was accompanied by a decrease in the compression strain; therefore, at an ultimate load equal to 44 kN, the c/d ratio was 0.29.

The BT4, BT5, and BT6 specimens behaved similarly to the reference specimen in the elastic stage. However, the presence of the openings led to the loss of a part of the strength and decreased the specimens’ strain-hardening capacity. This affected the type of failure in the compression part. Therefore, the presence of openings in these specimens led to buckling at the openings without yielding.

#### 3.6.3. The Third Group

The strain distribution at the top and bottom surface for the BT7 and BT8 specimens is shown in Figure 34 and Figure 35. Focusing on Figure 34, it can be noted that the BT7 specimen showed a very similar behavior at both the top and bottom surface under loading. This behavior is represented by a linear relationship between the applied load and the resulting strain. Therefore, the N.A. location remained constant from the beginning of loading up to a load equal to 48 kN, as shown in Figure 36, because of the equal increase in both compression and tension strain.

However, increasing the applied loads increased the compression strain values compared with the tensile strain, and it caused a decent in the N.A. location to the bottom of the cross-section center and increased the depth of the compression zone. At the ultimate load, the tension strain value was reduced by 8.83% compared to the compression strain. Meanwhile, the gradual loading of the BT8 specimen led to the strain values of the top surface being larger than those of the bottom surface, as shown in Figure 36; therefore, the N.A. location moved to the bottom of the specimen and increased the depth of the compression zone at loads from 10 to 25 kN, as shown in Figure 36. After these loads, the N.A. location and the depth of the compression zone remained constant in the bottom of the specimen because of the equal increase in both compression and tension strain.

When the specimen reached the yield load, it was observed that the strain value of the top surface became larger than that of the bottom surface. Therefore, the N.A. location descended more than in the previous stages, and the depth of the compression zone increased more significantly. After this, this situation was prolonged due to the stress redistribution, which significantly increased the compression strain. The tension strain was 66.79% lower at the ultimate load than compression strain.

From the above, it can be noted that increasing the specimen length served to increase the compatibility between the tension strain at the bottom surface and the compression strain at the top surface and led to the N.A. location remaining constant from the beginning of loading to approximately the ultimate load, as in the BT7 specimen. Meanwhile, decreasing the length of the BT8 specimen increased the exposure of the top surface to compression strain more significantly compared with the tensile strain, and thus the N.A. location continued to descend until the loading end.

#### 3.6.4. The Fourth Group

Figure 37 and Figure 38 show the top and bottom strain distribution for the BT9 and BT10 specimens, respectively. Figure 37 shows the linear relationship between the applied load and the resulting strain for the BT9 specimen. This relationship continued to a load equal to 140 kN for the top surface strain and 120 kN for the bottom surface. The specimen loading led to different strain values at both surfaces. However, the values of the bottom surface were larger than the top and thus caused the continuation of the ascent of the N.A. location to the top of the specimen and decreased the depth of the compression zone at the ultimate load equal to 185 kN, as shown in Figure 37. At this load, the c/d ratio reached 0.176 because of the local buckling failure, which reduced the compression strain by 78.54% compared with the tensile strain. At the same time, the BT10 specimen showed a similar behavior at both the top and bottom surface. It is represented by a linear relationship between the vertical load and the strain at the elastic stage, as shown in Figure 38. The BT10 specimen showed approximately equal strain values at these loads, which caused the N.A. location to remain constant up to the yield load, as shown in Figure 39. After this, an increase in the tension strain occurred and caused the ascent of the N.A. location and decreased the compression zone’s depth. However, the ongoing loading process led to an increase in the compression strain values due to the stress redistribution for this specimen and thus caused the descent of the N.A. location and increased the depth of the compression zone.

Increasing the specimen’s diameter clearly affected the type of failure and the location of the N.A. through reducing the exposure of the top surface to compression strain more significantly compared with the tension strain at the bottom surface, and led to the appearance of buckling without the occurrence of yielding, as in the BT9 specimen. Meanwhile, decreasing the specimen’s diameter increased the compatibility between the compression and tension strain. In turn, it led to the steadiness of the N.A. location until the yield load occurred.

### 3.7. Strain Distribution at Loading Points

Application of the loads at the mid-span, far left, and far right points on the specimens led to equal strain and similar loading behavior, which could be characterized by a linear relationship between the applied load and the resulting strain up to the yield load. In this stage, the values of the strain at mid-span and the two loading points were very similar, indicating that the specimen deformation was minimal. After the yield point, the stresses redistributed because of the strain-hardening capacity of these specimens, which lead to a significant increase in the strain values at the mid-span compared with the two far loading points. Therefore, it led to the spacing of the strain values at the mid-span and the two loading points.

Finally, the present study included the preparation and testing of 10 circular hollow beam specimens up to and after the failure stage by using different sizes and with diameter-to-thickness ratios ranging from 16.93 to 73 under bending load.

## 4. Conclusions

Based on the observations and results obtained in this experimental work, bearing strength, stiffness, and ductility increased significantly (81.75%, 22.66%, and 58.04%, respectively) on increasing the section thickness by 200% in hollow steel specimens (without openings). Meanwhile, a decrease in bearing strength, stiffness, and ductility was observed of 38.87%, 49.21%, and 64.86%, respectively, on decreasing the section thickness by 33.33%. An increment of 115.55% in specimen diameter leads to a corresponding increase in the bearing strength and stiffness by 237.59% and 221.56%, respectively; however, it reduces the specimen ductility by 76.67% and affects the failure mode. On reducing the specimen diameter by 25%, we noted a decrement in the bearing strength and stiffness by 56.75% and 76.61%, respectively.

On increasing the span of specimens by 33.33%, all the parameters observed, i.e., bearing strength, stiffness, and ductility, decreased by 3.28%, 58.34%, and 37.13%, respectively. However, with the reduction in span by 33.33%, bearing strength and stiffness were observed to be increased by 89.78% and 58.59%, respectively, but the ductility was reduced by 50.21%, changing the failure mode.

The provision of square openings along the span of specimens significantly impacted the structural parameters considered. Bearing capacity was reduced by 17.88%, 19.71%, and 14.23% and ductility was reduced by 72.40%, 67.71%, and 60.88% on providing one, two, and three openings, respectively. The same openings abridged the stiffness by 39.83%, 20.22%, and 41.90%, respectively.

Varying the location of a single square opening along the span of the reference specimen significantly affected the ultimate strength, and the location of the opening at 30%, 40%, and 50% of the reference specimen’s length reduced the ultimate strength by 8.76%, 14.23%, and 17.88%, respectively.

The local buckling failure dominated for specimens having a D/t ratio more than 50 and showed very negligible levels of ovalization of the cross-section. Local buckling failure was observed to be prevented after providing circular rings in the specimen, since bearing strength was increased by 53.24% compared with the specimen without rings.

## Data Availability

All data are available in the paper.

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
