# Peer review of "Experimental Analysis of Steel Circular Hollow Section under Bending Loads: Comprehensive Study of Mechanical Performance"

_materials, 2022, doi:10.3390/ma15124350_

Round 1
Reviewer 1 Report
Dear Khalaf et al.,
The manuscript “Experimental Analysis of Steel Circular Hollow Section Under Bending Loads: Comprehensive Study of Mechanical Performance” (materials-1744854) by Khalaf et al. aimed at evaluating the mechanical performance under bending 25 loads of circular hollow sections of steel. The topic is interesting, but I think this article should reconsider after proper changes in major revision for publication in Materials. Some of my specific comments are below:
- The abstract (line 25-45) is poorly written due to not being concise, please rewrite it with a more solid explanation.
- Describe the novelty of the article made by the author? From the results of my evaluation, it seems that many similar published works adequately explain what you have raised in the current manuscript related to circular steel on bending testing based on the reviewer's expertise and knowledge in this research area. If there are something others really new in this manuscript, please highlight it more clearly in the introduction section (line 49-95).
- The state of the art and the significance of the current study are not clearly present, the authors should highlight it more advanced in the introduction section (line 49-95).
- In the introduction section (line 49-95), the authors should explain the previous research conducted and its shortcomings. It will uphold the research gap that you filled with your research novelty. I recommend the authors elaborate on their introduction section. Do not forget to attention carefully to my previous comments on numbers 2 and 3.
- Since this manuscript evaluates the mechanical performance of metallic materials, I would encourage and advise the authors to adopt some of the specific additional references related to metallic material evaluation based on mechanical performance published by MDPI in the introduction section (line 49-95) as follow:
-
- Tresca Stress Simulation of Metal-on-Metal Total Hip Arthroplasty during Normal Walking Activity. Materials (Basel). 2021, 14, 7554. https://doi.org/10.3390/ma14247554
- The Effect of Bottom Profile Dimples on the Femoral Head on Wear in Metal-on-Metal Total Hip Arthroplasty. Journal of Functional Biomaterials. 2021, 12, 38. https://doi.org/10.3390/jfb12020038
- In the materials and methods section (line 86-130), the authors should add one systematic figure to illustrate the workflow of experimental testing in the present study to make the reader more interested and easier to understand rather than only using dominant text to explain.
- It is crucial to explain more clearly why bending loads needs to be investigated on Steel hollow as conducted in the present study, why not axial and/or torsion loads?.
- The author must provide a detailed specification and use condition more detail regarding all tools used in the research carried out so that the reader can estimate the accuracy and differences in the results that the authors describe due to the use of different tools in future studies.
- In the Results and discussion section (line 239-899), the authors are advised to compare the results they obtain with previous similar/identical studies if it is possible.
- In the last paragraph before the conclusion section (after line 899), the authors should add of one paragraph about the limitations of the presented study.
- The conclusion (line 900-931) of the present manuscript is not solid. Further elaboration is needed. Also, make it into paragraph, not point-by-point as in present form.
- Further research needs to be explained in the conclusion section (line 900-931).
- In the whole of the manuscript, the authors sometimes made a paragraph only consisting of one or two sentences that made the explanation not clearly understood. The authors need to extend their explanation to become a more comprehensive paragraph. In one paragraph, it is recommended to consist of at least 3 sentences with 1 sentence as the main sentence and the other sentences as supporting sentences. For example in line 218-221.
- I see some errors on English in some areas of the present manuscript. To improve the quality of English used in this manuscript and make sure English language, grammar, punctuation, spelling, and overall style are correct, further proofreading is needed. As an alternative, the authors can use the MDPI English proofreading service for this issue.
- Please make sure the authors have used the Materials, MDPI format correctly. The authors can download published manuscripts by Materials, MDPI, and compare them with the present author's manuscript to ensure typesetting is appropriate. For example line 57-58.
-
- Email from all of the authors should whitten as black color with initial in the end of the email if in the one affiliation have two or more authors
- The way to present figure and table is not correct
- And other
I am pleased to have been able to review the author's present manuscript. Hopefully, the author can revise the current manuscript as well as possible so that it becomes even better. Good luck for the author's work and effort.
Best regards,
The Reviewer
Reviewer 2 Report
Dear Authors, I read your article submitted in MDPI-Materials. Below are my comments.
TITLE AND ABSTRACT
3. The title of the manuscript conveys with the major concern of the study.
4. The abstract properly summarize the topic addressed, but it is too wordy. Please synthetize.
INTRODUCTION
5. The null hypothesis is missing at the end of the introduction section. Please add.
MATERIALS AND METHODS
6. The full chemical composition of the steel specimens used should be given.
7. How did you decide sample size? Did you take it from previous study or did you perform a sample size analysis? Please clarify. A sample size calculation is always preferred.
RESULTS
8. I do no get why for quantitative data you did not perform the statistical analysis. Giving percentage is fine, but more sample should be investigated in my opinion and check for significant result, in order to gain scientific soundness.
REFERENCES
9. Please check to have correctly numbered all the references. Order them correctly in the main text.
Round 2
Reviewer 1 Report
Good works from the authors, I recommend this manuscript to accepted for publication.
Reviewer 2 Report
No any other comments